# PhyloQuant approach provides insights into *Trypanosoma cruzi* evolution using a systems-wide mass spectrometry-based quantitative protein profile

Simon Ngao Mule[1], André Guilherme Costa-Martins[1], Livia Rosa-Fernandes [1], Gilberto Santos de Oliveira [1], Carla Monadeli F. Rodrigues[1], Daniel Quina[1], Graziella E. Rosein[2], Marta Maria Geraldes Teixeira[1] & Giuseppe Palmisano[1✉]

The etiological agent of Chagas disease, *Trypanosoma cruzi*, is a complex of seven genetic subdivisions termed discrete typing units (DTUs), TcI-TcVI and Tcbat. The relevance of *T. cruzi* genetic diversity to the variable clinical course of the disease, virulence, pathogenicity, drug resistance, transmission cycles and ecological distribution requires understanding the parasite origin and population structure. In this study, we introduce the PhyloQuant approach to infer the evolutionary relationships between organisms based on differential mass spectrometry-based quantitative features. In particular, large scale quantitative bottom-up proteomics features (MS1, iBAQ and LFQ) were analyzed using maximum parsimony, showing a correlation between *T. cruzi* DTUs and closely related trypanosomes' protein expression and sequence-based clustering. Character mapping enabled the identification of synapomorphies, herein the proteins and their respective expression profiles that differentiate *T. cruzi* DTUs and trypanosome species. The distance matrices based on phylogenetics and PhyloQuant clustering showed statistically significant correlation highlighting the complementarity between the two strategies. Moreover, PhyloQuant allows the identification of differentially regulated and strain/DTU/species-specific proteins, and has potential application in the identification of specific biomarkers and candidate therapeutic targets.

[1] Department of Parasitology, Institute of Biomedical Sciences, University of São Paulo, São Paulo, Brazil. [2] Department of Biochemistry, Institute of Chemistry, University of São Paulo, São Paulo, Brazil. ✉email: palmisano.gp@usp.br

*T*rypanosoma cruzi is a protistan parasite agent of Chagas disease (American trypanosomiasis), a zoonotic disease endemic in 21 Latin American countries[1] with ~25 million people at risk of infection. An estimated 6–7 million people are infected worldwide and 10,000 annual mortalities are reported[2,3]. The hematophagous triatomine insects (Reduviidae: Triatominae) are the main vectors of *T. cruzi*. Other transmission routes include blood transfusion[4–6], oral transmission from food or beverage contamination[7–10], congenital transmission[11,12] and organ and marrow transplant from infected donors[13,14]. Chagas disease is currently an emergent public health concern due to the outbreaks of oral infection, especially in the Amazon region[7–10], and international migrations contributing to many cases of the disease in non-endemic countries including Europe, USA, Australia and Japan[15,16].

During its life cycle, *T. cruzi* alternates between triatomine vectors and mammalian hosts[17]. Epimastigotes proliferate in the insect midgut and differentiate into metacyclic trypomastigotes, the non-proliferative stage that infects mammals through skin wounds or mucosal membranes, where they invade the cells and differentiate into amastigotes. The amastigotes replicate intracellularly and differentiate into infective trypomastigotes, which are released into the bloodstream following host cell rupture[18]. Each developmental stage is characterized by changes in morphology, molecular and biochemical makeup[17,19–23].

The variable clinical outcomes of Chagas disease have been associated to *T. cruzi* heterogeneity[24–26]. Different studies have elucidated the genetic variability in *T. cruzi* populations and shown close associations between the parasites' genetics and relevant biological, pathological and clinical characteristics[24,25,27–29]. In 1999, a consensus nomenclature was proposed, placing *T. cruzi* strains into two major groups; *T. cruzi* I and *T. cruzi* II[30]. In 2009, this classification was revised, distributing. *T. cruzi* strains into six genetic subdivisions termed discrete typing units (DTUs)[31]. Currently, seven DTUs are recognized in *T. cruzi* (TcI-TcVI)[32], along with Tcbat, which infects predominantly bats[33–35]. The parasite's DTU has been implicated as one relevant factor influencing clinical variations of the disease[24,27,36], drug resistance/susceptibility[37,38], pathogenicity in mice[38], and vector competence[18,39]. Other factors influencing disease outcome include mixed infections, infection routes, host genetics and a range of eco-geographical factors[32,40,41]. Different molecular techniques have been used to study the genetic diversity of *T. cruzi*, such as kinetoplast DNA[42], RAPDs[43,44], polymorphism of the spliced leader intergenic regio[45–48] and multilocus sequence typing[49–52].

The application of mass spectrometry in the typing of organisms has been previously described[53,54]. Moreover, the comparison between proteomes of *C. elegans* and *D. melanogaster* showed that protein abundance between organisms is more comparable than mRNA transcripts and more similar than protein abundance and transcript expression within the same organism[55]. Telleria et al. have described the hierarchical clustering of *T. cruzi* strains using Euclidean Distances based on protein expression levels from 2D-DIGE analysis[56]. This clustering showed a high correlation between *T. cruzi* phylogenetics and levels of protein expression. In addition, *T. cruzi* strains have been assigned to their DTUs with high accuracy based on the genome-free spectral matching assay[57]. In the current study, we applied a proteome-wide quantitative proteomics approach to profile and infer evolutionary relationships between *T. cruzi* DTUs (TcI-TcVI, Tcbat) and their most closely related bat trypanosomes; *Trypanosoma cruzi marinkellei*, *Trypanosoma dionisii*, and *Trypanosoma erneyi*, all classified in the subgenus *Schizotrypanum* nested into *T. cruzi* clade[33,35,58–60]. The generalist and human-infective *T. rangeli* also belonging to the *T. cruzi* clade was included as an outgroup of *Schizotrypanum*

subgenus[59,61]. This approach was termed "PhyloQuant", and is herein introduced to encompass phyloproteomics approaches with other identified and unidentified biomolecular features obtained by quantitative proteomics, metabolomics, lipidomics, glycoproteomics and other omics techniques in typing and evolutionary inferences of organisms. Genetic distance matrices from PhyloQuant-based phylogenies compared to genetic distance matrices from phylogenies inferred from concatenated SSU rRNA, glycosomal glyceraldehyde 3-phosphate dehydrogenase (gGAPDH) and HSP70 gene sequences were analyzed to determine their levels of correlation. We show the strong discriminatory power of three MS-based quantitative features to accurately assign *T. cruzi* strains into their DTUs, and the suitability of this technique to reconstruct the evolutionary relationships among the DTUs of *T. cruzi*, and between this species and closely related trypanosomes. Our findings support a close congruence of the evolutionary relationships among all assayed trypanosome species and *T. cruzi* DTUs using PhyloQuant and sequence-based phylogenetic studies. Putative *T. cruzi* strain/DTU, and species-specific proteins at the exponential growth phase of the epimastigote life stage are also explored.

## Results

**Mass spectrometry results**. The evolutionary relationships of *T. cruzi* strains representative of all DTUs and closely related trypanosomes of the subgenus *Schizotrypanum* were investigated using a comprehensive bottom-up mass spectrometry-based quantitative proteomics combined with statistical and computational analyses. A schematic flowchart of the experimental procedures applied in this study is summarized in Fig. 1. A total of 524,303 MS1 values from precursor ions with a charge state of ≥2 were quantified from the 17 trypanosomes in all the biological replicates evaluated. Of these, 68,828 precursor ion intensities had a minimum of three valid values in at least one group, of which 52,622 were statistically regulated using ANOVA (Supplementary Data 1). A total of 4796 proteins were identified and quantified in this study (Supplementary Data 2). Of these, 2441 and 4018 label-free quantification (LFQ) and intensity-based absolute quantification (iBAQ) values, respectively, were statistically significant (FDR < 0.05), with three valid values in at least one group (Supplementary Data 2 and 3, respectively). All the quantitative proteomic values from total and statistically significant MS1, iBAQ and LFQ features were normalized by z score, and maximum parsimony (MP), a phylogenetic algorithm, used to infer evolutionary relationships.

**Proteome variation among trypanosomes**. A multivariate data analysis based on principal component analysis (PCA) was employed to explore the variation in protein expression between trypanosomes obtained by LC-MS/MS. Variations of Log2-transformed MS1, iBAQ and LFQ were visualized by the first and second principal components (Fig. 2). PCA on MS1 intensities placed the *T. cruzi* DTUs into a non-cohesive assemblage comprising the highly phylogenetically related *T. c. marinkellei*, whereas the more distant related *T. dionisii*, *T. erneyi* and *T. rangeli* clustered separately (Fig. 2a). This clustering was based on the first and the second principal components, which represented 11% and 7% of the total variation in the dataset, respectively. Proteome variation based on iBAQ values resulted in a much more resolved clustering (Fig. 2b), showing the separation of *T. cruzi* DTUs from all allied bat trypanosomes. Clustering analysis of *T. cruzi* DTUs by PCA showed that TcI and Tcbat clustered tightly together closest to a cluster exclusive of TcIV strains based on the second principal component representing 10% of the total variation in the dataset. Strains representative of

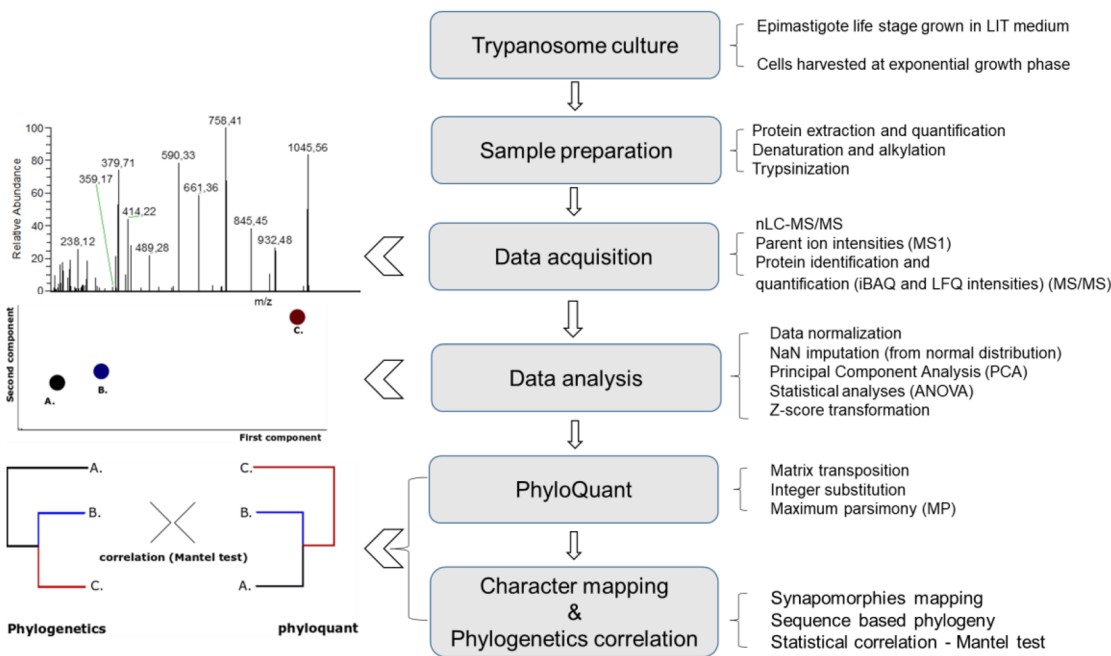

**Fig. 1 Experimental workflow of the PhyloQuant approach.** Trypanosomes grown in complete LIT medium were harvested at the exponential proliferative phase of the epimastigote life stage. Proteins were extracted in 8 M urea lysis buffer in the presence of protease inhibitors, digested with trypsin and the resultant peptides analyzed by nLC-MS/MS. Normalized MS1, iBAQ and LFQ intensities were used to infer evolutionary relationships using the Maximum Parsimony criterion, implemented in the TNT program. Character mapping of synapomorphies that differentiate each clade was performed in TNT program. Phylogeny based on three concatenated genes, gGAPDH, V7V8 SSU rRNA and HSP70 was chosen to perform correlation of the evolutionary relationships based on proteome-wide mass spectrometry-based features. Statistical levels of correlation between PhyloQuant and phylogenetic distance matrices were inferred using the Mantel test.

TcIII clustered very near to TcII forming a cohesive cluster also comprising strains of TcV and TcVI. *T. c. marinkellei* was placed well separated from both *T. cruzi* DTUs and the other bat trypanosomes *T. dionisii* and *T. erneyi*. *T. rangeli* was placed furthest from all *Schizotrypanum* trypanosomes based on the first principal component which represented 30.9% of the total variation in the dataset.

PCA analysis based on LFQ intensities resulted in four distant clusters based on the first and second components representing 35.4% and 10.9% of the total variability in the dataset, respectively (Fig. 2c). All *T. cruzi* DTUs clustered closely together, although TcI, Tcbat and TcIV were placed at the edge of the major cluster comprising all *T. cruzi* strains, and the parental genotypes TcII and TcIII and their respective hybrid genotypes TcV and TcVI[27–29,32] formed a very tight cluster. Similar to iBAQ based PCA, *T. c. marinkellei* was clearly separated from *T. cruzi* DTUs, *T. dionisii* and *T. erneyi*.

**Protein expression-based evolutionary inference ('Phylo-Quant').** In this proof of concept study, we evaluated the potential application of quantitative proteomic-based data for DTU assignment and inference of evolutionary relationships of *T. cruzi* strains and closely phylogenetically related trypanosomes. Phylogenies were inferred by MP using numeric values from total and statistically significant unidentified precursor ion (m/z) intensities; and total and statistically significant intensities of identified MS-based proteomic features (iBAQ and LFQ intensities). Subsequently, correlation of the PhyloQuant approach was performed against phylogenies inferred from concatenated sequences of three gene loci; gGAPDH, V7V8 SSU rRNA and HSP70 (Supplementary Fig. 1). Character mapping based on statistically significant LFQ-based PhyloQuant were mapped, showing the proteins and their expression profiles which support

the different monophyletic clades (Supplementary Data 4). PhyloQuant clustering inferred from statistically significant MS1 m/z intensities showed the separation of all *T. cruzi* DTUs including Tcbat from the closely phylogenetically related trypanosomes strongly linked to bats; *T. c. marinkellei*, and the sister clade comprising *T. erneyi* and *T. dionisii* (Fig. 3a). *T. c. marinkellei* formed the basal clade to *T. cruzi* DTUs, supported by a high bootstrap value. *T. cruzi* strains were grouped in two major clusters; one composed of TcI-Tcbat/IV/III strains, and the second composed of TcII/III/V/VI. Strains representative of TcI, TcIV and TcVI clustered to form monophyletic clades. TcII strains and one strain of TcIII clustered with those of TcV and TcVI, consistent with TcII and TcIII hybridization events given origin to TcV and TcVI. In addition, corroborating relevant intra-DTU TcIII genetic diversity, the TcIII strain M-6241 clustered with those of TcIV. Tcbat was placed sister to TcI forming a clustering congruent with evolutionary relationships inferred by various genetic markers including cytb, gGAPDH, SSU rRNA, H2B and many other gene sequences[33–35]. PhyloQuant clustering based on total MS1 intensities strongly supported the separation of *T. cruzi* DTUs from their allied trypanosome species, including *T. c. marinkellei*, but showed less inter DTU discriminatory power (Supplementary Fig. 2a). Although only TcIV and TcVI strains formed DTU-specific clades, the analysis showed the expected close relationships among TcII, TcIII, TcV and TcVI.

PhyloQuant clustering based on the identified MS features, i.e., statistically significant iBAQ and LFQ intensities (Fig. 3b, c), and total iBAQ and LFQ intensities (Supplementary Fig. 2b, c), showed very well resolved clustering of the *T. cruzi* strains and closely related trypanosomes with higher bootstrap support values. PhyloQuant based on these features supported with high bootstrap values the separation of all *T. cruzi* DTUs from *T. c. marinkellei*, *T. dionisii* and *T. erneyi* and the outgroup *T. rangeli*. Notable, these quantitative features showed a high inter DTU

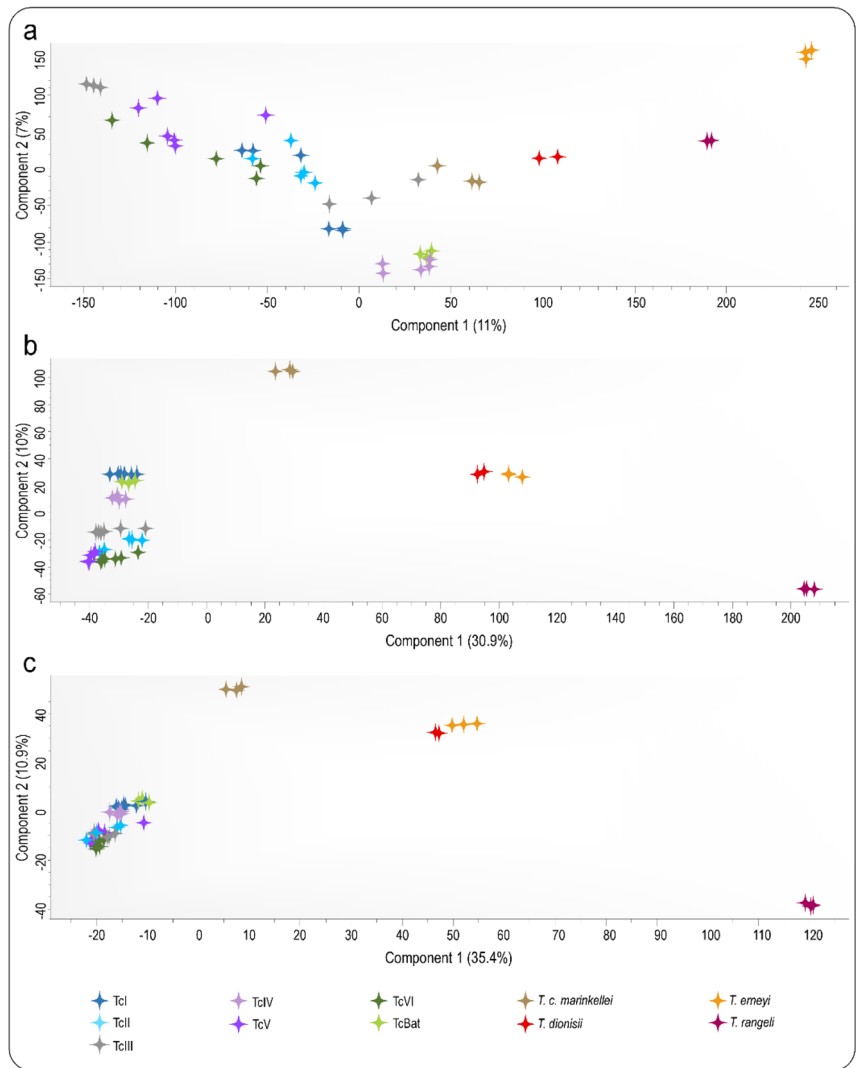

**Fig. 2 Principal component analysis of *T. cruzi* DTUs and bat allied trypanosomes; *T. c. marinkellei*, *T. dionisii*, *T. erneyi*, and the generalist *T. rangeli*.**
The first and second principal components based on normalized **a** MS1 values, **b** iBAQ and **c** LFQ proteomic features representing the total proteome variations in the three datasets are presented.

discriminatory power, enabling the assignment of *T. cruzi* strains to their DTU clades, with the exception of TcII strains, which remained intermingled with TcV or TcVI. Three main clusters were formed with *T. cruzi* strains of the seven DTUs; TcI-Tcbat were placed as sister clades, and formed the basal clade to *T. cruzi* DTUs. TcIV strains clustered separated between the strongly supported clades TcI-Tcbat and TcIII. The parental TcIII was basal to a major clade harboring monophyletic TcV and TcVI strains, and the paraphyletic TcII. Corroborating different strains given origin hybrid DTUs, *T. cruzi* Esmeraldo was placed closer to TcV, while *T. cruzi* Y showed a close relationship with TcVI. Phylogenetic character mapping of the proteins supporting each clade (synapomorphies), and their respective expression profiles, were determined using the TNT program, and are presented in Supplementary Data 4.

**PhyloQuant shows high correlation with phylogenetics**. To infer evolutionary correlation between PhyloQuant and phylogenetics, the replicate *z* score values of each strain were concatenated and used to infer PhyloQuant-based trees based on statistically significant and total MS1, iBAQ and LFQ intensities.

Subsequently, distance matrices from PhyloQuant and molecular-based evolutionary trees inferred by MP based on concatenated gGAPDH, SSU rRNA and HSP70 gene sequences (Supplementary Fig. 1) were analyzed for levels of congruency. Phylogenies inferred by the concatenated DNA sequences have previously been employed to infer phylogenies aimed at resolving relationships between trypanosome species of the subgenus *Schizotrypanum* lineage and the Tra [Tve-Tco] lineage[61] comprising *T. rangeli*, *T. conorhini*, *T. vespertilionis* and other Old World trypanosomes.

Levels of correlation were statistically evaluated using the nonparametric Mantel test using analyses of phylogenetics, implemented in R statistical software. Mantel test gave a statistically significant correlation (*p* value of 0.0009), showing close correlation between three evolutionary trees based on MS1, iBAQ and LFQ and phylogenetics. Levels of correlation were determined based on 1000 permutations. The statistically significant correlation between phylogenies derived from protein expression levels and molecular-based phylogenies illustrate the high congruence between *T. cruzi* protein expression and infraspecific genetic subdivisions, as earlier illustrated by Telleria et al.[56].

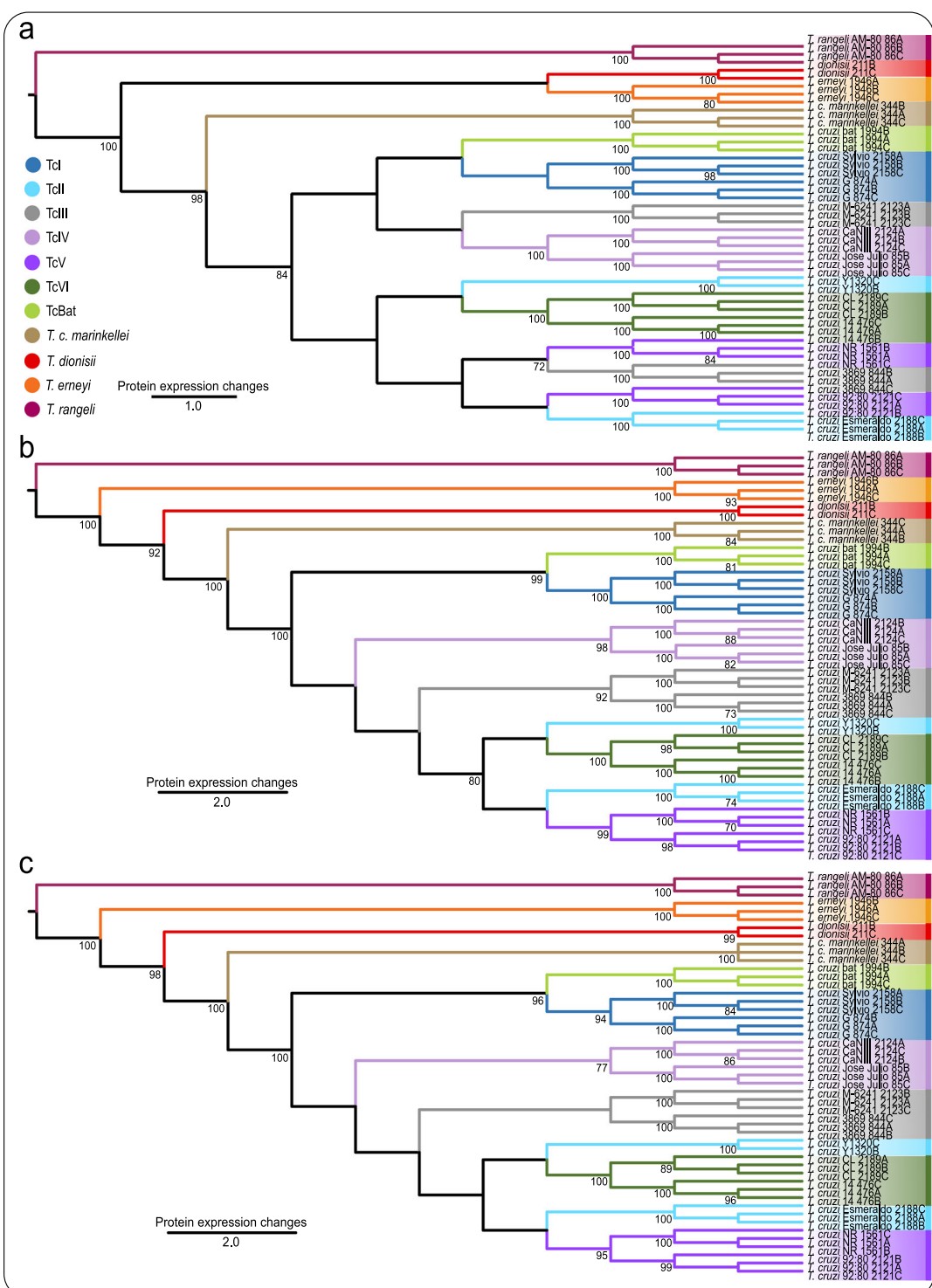

**Fig. 3 PhyloQuant analysis of trypanosomes of the *Schizotrypanum* subgenus based on statistically regulated MS features. a** MS1, **b** iBAQ and **c** LFQ values inferred by the parsimony evolutionary algorithm. Numbers on branches represent bootstrap support above 70% estimated with 1000 replicates.

**DTU and species-specific proteins**. Species, DTU and strain exclusive proteins were identified in this study. It should be noted that the identification of exclusive proteins does not imply the absence of the protein in other groups, but can be related to the abundance below the limit of analytical detection, the presence of modified peptides and/or unique peptide sequences. The number of strain/species and DTU-specific proteins of the human-infective strains is summarized in Fig. 4a, and the proteins and the number of unique peptides for each protein is presented in

Supplementary Data 5. Among the human-infective *T. cruzi* strains, *T. cruzi* 3869, a TcIII strain, had the highest numbers of exclusively identified proteins, of which nine were assigned to single proteins. Of these, four proteins were assigned to the multigene family retrotransposon hot spot (RHS) protein (Q4D2F5, Q4DM62, V5AUD1 and V5BBB7). *Tcbat*TCC1994, a representative strain of the bat restricted *T. cruzi* DTU, had 17 exclusive proteins, of which seven were assigned to single proteins including chaperonin HSP60/CNP60, putative (Fragment)

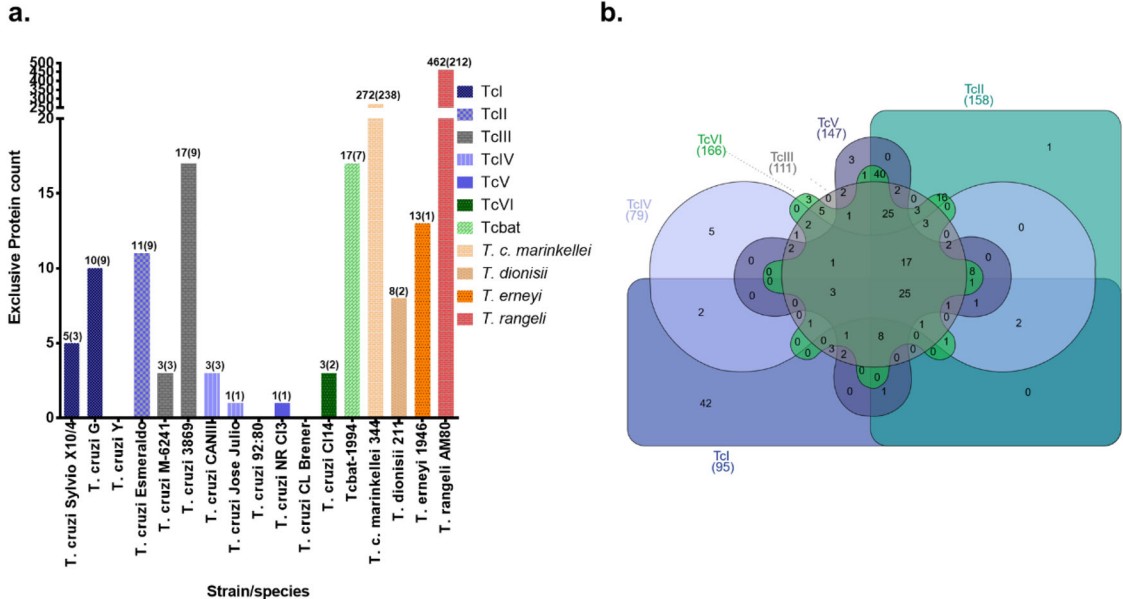

**Fig. 4 Analysis of strain/DTU and species-specific proteins identified and quantified by nLC-MS/MS analysis of epimastigote stage parasites at the exponential growth phase. a** The number of proteins identified exclusively for each *T. cruzi* strain and closely related trypanosome species, (proteins identified exclusively and not in protein groups are highlighted in parenthesis). **b** Venn diagram of DTU-specific proteins identified and quantified in this study.

(K2MB79) and a putative vesicle-associated membrane protein (K2MYF8). In addition, three trans-sialidase protein groups were identified exclusively to Tcbat. *T. c. marinkellei* and *T. rangeli* had the highest numbers of exclusive proteins, with 272 and 462, respectively. On the other hand, a total of 8 and 13 proteins were identified and quantified exclusive for *T. dionisii* and *T. erneyi*, respectively (Supplementary Data 5). The analysis of exclusive human-infective DTU-specific proteins revealed varying numbers of proteins exclusively identified as DTU-specific proteins/protein groups (Fig. 4b). TcI had the highest number (42) of exclusively identified DTU-specific proteins (Supplementary Data 5). One protein, FAD-binding PCMH-type domain-containing protein (Q4DPS1), was identified exclusive to TcII with three unique peptide counts, while no TcIII exclusive protein(s) was identified based on a cutoff of five valid values per DTU. TcIV had five exclusive proteins, including two RHS proteins (Q4DII2/Q4DMF9 and V5B362) and a putative Thiol-dependent reductase 1 (K2N3V3) identified with 17 unique peptides. All three proteins identified exclusive to TcV were putative RHS proteins (Q4DBD1, Q4CKU7 and Q4CU66), while three proteins were identified exclusive to TcVI; RHS protein group (Q4CQ60/V5B746/V5B8V7/V5B362), a putative Hsc70-interacting protein (Hip) (Q4DAT6) and an uncharacterized protein (Q4E501/A0A061IZW3).

## Discussion

In this study, an approach to typing and evolutionary inference between organisms based on systems-wide protein expression profiles is described, and was applied to trypanosomes of the subgenus *Schizotrypanum*, a monophyletic assemblage nested in the clade *T. cruzi* comprising mostly trypanosomes of bats, and two species infective to mammals in general, including humans: the pathogenic *T. cruzi* and the non-pathogenic *T. rangeli*[34,35,59–62]. This approach is based on different biomolecular features obtained by bottom-up large scale mass spectrometry-based quantitative proteomics analysis (MS1, iBAQ and LFQ) followed by MP, an evolutionary criterion to infer evolutionary distances between

strains/species. This analysis requires a nomenclature that describes evolutionary inferences of species or strains using quantitative mass spectrometry data. We propose the term "PhyloQuant" to indicate methodologies which use biomolecular features obtained by mass spectrometry-based methods to infer evolutionary relationships between organisms. Historically, evolutionary relationships are deduced from morphology, or DNA/RNA and predicted protein sequence data based respectively on phenotype, or nucleotide and amino acid sequences from different organisms for their comparisons. Evolutionary trees build using phylogenetic approaches use MP[63], maximum likelihood (ML)[64] and Bayesian inference (BI) methods[65]. PhyloQuant approach uses MP to infer evolutionary relationships between organisms. MP uses the lowest number of evolving steps as a criterion to drive the search for the evolutionary trees, and does not imply the use of an evolutionary model for clustering, which are required for ML and BI inferences. In addition, MP can track changes of a character (synapomorphies), herein differences in expression of different proteins that support each clade. The proteins that support the different trypanosome clades and their differential expression profiles based on LFQ intensities, are shown in Supplementary Data 4. Phyloproteomics compare biological clustering using phylogenetic algorithms based on mass spectrometry protein/peptide peaks appearance or disappearance. Abu-Asad et al. used the term phyloproteomics to classify serum cancer and non-cancerous samples using parsing and a phylogenetic algorithm[66]. In particular, SELDI-TOF MS data were obtained from serum collected from cancer and non-cancer patients and the m/z values were sorted for polarity in order to identify derived vs ancestral peaks using MIX, a phylogenetic algorithm. Another study investigated *C. jejuni* to develop a phyloproteomic typing scheme that combines the analysis of variable masses observed during intact cell MALDI-TOF MS with ribosomal MLST and whole genome MLST database-deduced isoform lists[67]. Another study used the term phyloproteomics to infer evolutionary relationships of bacteria based on mass spectrometry-based identification and protein sequence information[68]. The term PhyloQuant proposed in the present study unifies any method that uses quantitative omics data derived from

mass spectrometry to infer evolutionary relationships. It is important to note that the correspondence between any PhyloQuant and phylogenetic evolutionary relationships should be evaluated in each organism under study, and differences or similarities arising from this comparison holds important evolutionary and/or biological relevance.

PCA based on quantitative MS1 data clustered *T. c. marinkellei* closer to the major clade comprising all *T. cruzi* strains, while PCA based on iBAQ and LFQ intensities placed *T. c. marinkellei* between the clearly separated *T. cruzi* DTUs and the rest of the closely related trypanosome species (Fig. 2). Full genome analyses of generalist *T. cruzi* and *T. c. marinkellei* of bats revealed that they share majority of the core genes, and that *T. c. marinkellei* is also able to invade non-bat cells[60,69], including human cells, in vitro. However, although infection with *T. c. marinkellei* appears to be highly restrict to bat, and are unable of infecting mice[70], differences among the two genomes were mainly a reduced number of repetitive sequences in *T. c. marinkellei* genome compared to *T. cruzi*, with overall small nucleotide polymorphisms, and the identification of several putative exclusive gene sequences including 11 TcMU-CII and an acetyltransferase gene in *T. c. marinkellei* absent in *T. cruzi* Sylvio X10 and CL Brener genomes[69]. *T. c. marinkellei* has been placed in many phylogenetic studies as an outgroup to *T. cruzi* DTUs[34,59–61,69–71]. Phylogenies inferred from multiple genes on bat trypanosomes clustered *T. cruzi*, *T. c. marinkellei* and the cosmopolitan *T. dionisii* in the *Schizotrypanum* subgenus[34,59–61,70], which also included *T. erneyi*, a bat restricted trypanosome placed closest to *T. c. marinkellei*[60]. *T. rangeli*, included as the outgroup for *Schizotrypanum* species, has a wide range of mammalian hosts, which are shared with *T. cruzi* including man, but are not pathogenic to mammalian hosts. Mixed infections of *T. rangeli* with *T. cruzi* have been reported in humans and other mammals, as well as in their shared triatomine vectors[61,72]. *T. rangeli* belongs to the Tra [Tve-Tco] lineage, which is constituted by trypanosomes unable to develop intracellularly in vitro of unknown life cycle in the mammalian hosts, and is more closely related to Old World *T. vespertilionis* of bats and *T. conorhini* of rats (infective to non-human primates) than to the *Schizotrypanum* species[61,62]. Based on our study, PCA analysis based on quantitative protein expression datasets from MS1, iBAQ and LFQ quantitative features showed congruence to well-established evolutionary distances separating *T. cruzi* from *T. c. marinkellei*, *T. dionisii*, *T. erneyi* and *T. rangeli*[60–62].

Evolutionary relationships based on all three quantitative mass spectrometry features showed that *T. erneyi*, *T. dionisii* and *T. c. marinkellei* diverged earlier compared to *T. cruzi* DTUs[60–62,69,70]. *T. c. marinkellei* formed the basal clade to *T. cruzi* DTUs, supported by high bootstrap support values in all analyses (Fig. 3 and Supplementary Fig. 2). High levels of correlation between PhyloQuant and phylogenetic distance matrices were evidenced by Mantel test, and are in agreement with findings by Telleria et al.[56] who demonstrated the high correlation between gene expression, as shown by proteomic analysis using 2D-DIGE, and phylogenetic diversity assayed by multilocus enzyme electrophoresis. Our study demonstrates the potential application of quantitative systems-wide proteomics as a complementary method for taxonomic assignment of *T. cruzi* DTUs and closely related trypanosomes, and provides a valuable link between phylogenetic information and large- scale protein expression in the highly diverse *T. cruzi* strains.

The clustering of all trypanosome replicates in the same clades highlighted the reproducibility of our procedure. Compared to MS1 intensities, the evolutionary relationships of *T. cruzi* strains and closely related trypanosome species based on iBAQ and LFQ values showed the highest correlation to evolutionary relationships of *T. cruzi* inferred from phylogenies based on multilocus sequence typing, the golden standard proposed for the genotyping of *T. cruzi* in population studies[32]. Using 32 different gene loci, both separately and in concatenation, Flores-Lopez and Machado[51] reported the paraphyletic nature of *T. cruzi* II strains (now TcII-TcVI). Based on MS1, iBAQ and LFQ intensities, our analyses are in agreement with the clustering of *T. cruzi* TcII-TcVI. Two major clades of *T. cruzi* were described by Flores-Lopez and Machado[51]; one with TcI, TcIV, TcIII, and one haplotype from each TcV and TcVI; and the second clade harboring TcII, and the other haplotypes from TcV and TcVI. This branching pattern was corroborated by multilocus analyses including gene sequences from the genome of Tcbat, which was strongly supported as the DTU more closely related to TcI[35]. In our analyses, statistically significant MS1 based clustering showed a topology with two major groups; one composed of TcI-Tcbat, TcIV, and one representative strain of TcIII, and the second clade composed of TcII, TcV and TcVI, and the second representative strain of TcIII. The close clustering of TcIII and TcI has also been shown based on nuclear, mitochondrial and satellite DNA sequence[35,50,51,73].

Two hypotheses have been proposed to explain the evolutionary relationships of *T. cruzi* strains and the origin of the hybrid lineages, TcV and TcVI[27,28,32,74,75]. Based on PhyloQuant analysis (Fig. 3 and Supplementary Fig. 2, respectively), the close relationship between the parental TcII (*T. cruzi* Esmeraldo and *T. cruzi* Y) and the hybrid TcV and TcVI, was corroborated, in congruence to the well-accepted evolutionary history of *T. cruzi* DTUs. Based on iBAQ and LFQ intensities, TcIII strains formed a clade basal to the clade comprised of parental (TcII) and hybrid (TcV and TcVI) genotypes. Based on MS1 clustering, however, TcIII strains cluster with TcIV and TcV (Fig. 3a). Alternative topologies for TcIII strains have been reported in phylogenetic analysis using different gene markers[35,50–52]. The close relationship of TcI, TcIII and TcIV has been proposed in the 'Two hybridization' model in which initially TcI and TcII hybridized and gave origin to TcIII and TcIV, and a putative hybrid TcIII hybridizes with TcII, with secondary events of hybridization giving origin to TcV and TcVI strains[32,73–75]. Contributing to clarifying the evolutionary history of DTUs, TcIII was not confirmed as the result of a hybridization event between TcI and TcII based on multiple gene sequences[50–52,61,71] and analysis of satellite DNA[73]. The placement of DTU TcIV strains in a single clade is supported by phylogenies based on different genetic markers[35,50,51,71], suggesting a possible independent origin of this DTU[73].

Phyloquant offers the opportunity to include in a phylogenetic clustering different protein features. This implies that genes exposed or not to selective pressure are jointly considered. Since differentially expressed proteins can be associated to genes exposed to selective pressure, and consequently poor selective markers, using total quantitative proteomic features should leverage this issue. Based on total and statistically regulated iBAQ and LFQ quantitative features, highly similar tree topologies were obtained (Fig. 3 and Supplementary Fig. 2). However, based on unidentified features, statistically significant MS1 features had better resolved trees. Indeed, this method offers the possibility to identify strain/species-specific quantitative markers for discrimination. In our analyses, a total of 4796 proteins were identified and quantified, of which 2733 proteins were identified with three valid values in at least one condition, and used for PhyloQuant analysis based on LFQ intensities. Of these, 2441 proteins were statistically regulated while 292 were not differentially regulated (~10.68%), meaning that the differentially expressed protein pool contributes to the majority of proteins resolving the

differences between the strains compared to the non-regulated proteins. Imputation of non-assigned numbers (NaN) was performed prior to statistical and PhyloQuant analysis. A mass spectrometry data-dependent acquisition method was used, in which precursor ions are selected in a stochastic manner creating missing values between replicates and samples. A way to overcome this limitation would be the use of data independent acquisition where all precursor ions within a certain m/z range will be selected and fragmented having a deeper coverage with less missing values.

Epimastigote stage trypanosomes in the *Schizotrypanum* clade are confined to the digestive tract of triatomine and cimicid bugs, where they replicate in the midgut, and differentiate into metacyclic trypomastigotes in the hindgut from where they are passed through feces contamination to mammalian hosts during a blood meal[33,60,70]. On the contrary, *T. rangeli* is transmitted via salivarian route. Epimastigotes pass from the triatomine gut to the haemocoel, and then invade the salivary glands from where metacyclic trypomastigotes are transferred to mammalian hosts by bite during a blood meal[61,72]. The identification of differentially regulated proteins between *T. cruzi*, other species of *Schizotrypanum* and *T. rangeli* in the invertebrate hosts, which are all obligate hematophagous hemipterans, could help elucidate differences in adaptations of the parasites to survive, proliferate, and differentiate exclusively in the gut (*Schizotrypanum*), or invade the hemolymph and then reach the salivary glands (*T. rangeli*). Different strains of *T. cruzi* develop in a wide range of species of several triatomine genera. However, Peterson et al.[76] showed significant variations in *Rhodnius prolixus*' development following infections with different TcI strains. Contrasting with the well-known reference strains of DTUs TcI-TcVI, which are all infective to a range of experimentally infected triatomine species despite differences in both infection and metacyclogenesis levels, Tcbat was incapable of infecting triatomines from laboratory colonies, and its vectors remain to be identified[32,33,35]. Also controversial is the exclusive transmission of the widespread *T. c marinkellei* by rare triatomines of the genus *Cavernicola*[69,70]. The cosmopolitan *T. dionisii* develops similarly to *T. cruzi*, but, apparently exclusively in bat-associated cimicid bugs, which most likely are also vectors of *T. erneyi*, a species so far restricted to bats from Africa, where triatomines are absent[59–62,70]. The vectors of *T. rangeli* are triatomines of the genus *Rhodnius*, and biogeographical studies have supported strong association between complexes of *Rhodnius* species and the ability of salivary glands invasion by distinct phylogenetic lineages of *T. rangeli*[61,72]. A proteome-wide characterization of membrane proteins (surfaceome) revealed *T. rangeli* and *T. cruzi*-specific membrane proteins, including Gim5a and Lanosterol 14α-demethylase, exclusively in the epimastigote stage, and gp63 and Group II trans-sialidases in both epimastigote and trypomastigote life stages[77]. In our analysis, proteins that support separation between *T. rangeli* and the *Schizotrypanum* subgenus were mapped (Supplementary Data 4) based on the statistically significant LFQ intensities. A total of 456 protein expression profiles supported the large divergence of *T. rangeli* from species of the *Schizotrypanum* clade. Differential protein expression between *T. cruzi* and its closest relative trypanosomes from bats (stercorarian) and *T. rangeli* (salivarian) forms a platform to gain new insights into parasite-vector selection, development in the gut, haemocoel and salivary glands, and the regulated processes involved in these interactions.

Proteins exclusively identified in TcI included glucose-regulated protein 78 (V5B942) (Supplementary Data 5). The grp-78 is a member of the HSP70 family in the endoplasmic reticulum, and has been shown to be highly immunogenic against sera from *T. cruzi* infected mice[78]. Cyclophilin (V5B9Y6), a peptidyl-prolyl cis/

trans isomerase enzyme with diverse biological functions, was exclusively identified in *T. cruzi* strain MT3869 (TcIII) as a protein family (V5B9Y6/Q4DJ19/Q4DM35), sharing high sequence similarity. In addition, the expression profile of cyclophilin (V5D7H4, Q4E4G0) was also identified among the synapomorphies which support TcIII clade, and Tcbat (Supplementary Data 4). Cyclophilin is involved in protein folding, stress response, immune modulation, regulation of parasite cell death, and signal transduction. In *T. cruzi*, the cyclophilin gene family is comprised of 15 paralogous genes[79]. The upregulation of a *T. cruzi* cyclophilin, TcCyP19, was demonstrated in benznidazole resistant populations[80,81]. DTU-specific RHS proteins for TcI (V5BCL4), TcIV (Q4DII2/Q4DMF9 and V5B362) and TcVI (Q4CQ60/V5B746/V5B8V7) were also identified, in addition to several strain-specific RHS (Supplementary Data 5). Furthermore, *T. cruzi* strain-specific and trypanosome species-specific multigene families including trans-sialidases were also identified. The strain-specific trans-sialidase identified included (K2MR99) exclusive to *T. cruzi* M-6241 (TcIII), and a trans-sialidase protein group (Q4D8B3/V5B8Y6/K2MQD1/K2LZT0) exclusive to *T. cruzi* cl14 strain (TcVI). Three Tcbat (Q4CZE6, Q4D095 and Q4E2A1), one *T. dionisii* (A0A422MS29) and four *T. rangeli* (A0A3R7LX62, A0A3R7M8Q8, A0A422MQ19 and A0A422MT64) exclusive trans-sialidases were also identified. From the total proteins identified and quantified in this study (Supplementary Data 2), multigene families included 122 RHS proteins, 37 trans-sialidases and 20 dispersed gene family protein 1. Furthermore, several multigene family protein expression profiles were identified as synapomorphies that differentiate *T. cruzi* DTUs and related trypanosome species (Supplementary Data 4). The identification of multigene families among the most represented proteins and among the mapped synapomorphies indicate their diversity in expression according to hosts, life stages and developmental niches. This diversity might represent a unique signature for each strain/DTU interacting with its vertebrate and invertebrate hosts in a differential manner. The multigene families constitute 50% of the whole genomes s of *T. cruzi* strains, and their content has been shown to be different in phylogenetically distant strains[82]. The differential expression profiles of multigene families in different DTUs and closely phylogenetically related trypanosomes might regulate the infectivity, host restriction, host immune response, intracellular proliferation and transmission[82].

Three putative trans-sialidases (Q4CZE6, Q4D095, Q4E2A1), proteins with exo-alpha-sialidase activity involved in cell-invasion and intracellular survival of *T. cruzi* with important roles in the pathogenesis of Chagas disease were identified exclusively in Tcbat. A study by Maeda et al.[83] reported the surface molecules of Tcbat shared with TcI and corroborated the infectivity of metacyclic trypomastigotes to human Hela cells and mice despite lower parasitemia compared to virulent CL and even to the low parasitemic *T. cruzi* G of TcI[33,84]. Gp82, involved in host cell invasion by the induction of lysosomal exocytosis, was shown to be expressed and highly conserved in Tcbat, CL and G strains[83]. Mixed infection of Tcbat and TcI has also been reported by DNA detection in a child from northwestern Colombia[84], and Tcbat DNA has also been identified in mummies from Atacama desert by molecular paleoepidemiological tools[85]. The identification of putative Tcbat proteins or differential protein expression profiles can help to understand the biology and host–parasite interaction of this atypical DTU of *T. cruzi*.

Superoxide dismutase was among the proteins identified as exclusive to *T. c. marinkellei* (K2MYD1) with two unique peptides from seven identified peptides, and *T. rangeli* (A0A3S5IRU3) with two unique peptides from a total of six peptides. Based on a 2DIGE coupled to MS/MS study by Telleria et al.[56], a superoxide dismutase (Q4DI29) was identified

**Table 1 T. cruzi strains and closely related trypanosome species analyzed in this study.**

| TCC | DTU | Trypanosome strain/species | Host | Locality state/country |
|---|---|---|---|---|
| 2158 | TcI | *T. cruzi* Sylvio X10/4 | *Homo sapiens* | Pará, Brazil |
| 874 | | *T. cruzi* G. mucurae | *Didelphis marsupialis* | Amazonas, Brazil |
| 1320 | TcII | *T. cruzi* Y | *Homo sapiens* | Rio Grande do Sul, Brazil |
| 2188 | | *T. cruzi* Esmeraldo | *Homo sapiens* | Bahia, Brazil |
| 2123 | TcIII | *T. cruzi* M-6241 cl6 | *Homo sapiens* | Pará, Brazil |
| 844 | | *T. cruzi* 3869 | *Homo sapiens* | Amazonas, Brazil |
| 2124 | TcIV | *T. cruzi* CANIII cl1 | *Homo sapiens* | Para, Brazil |
| 85 | | *T. cruzi* Jose Julio | *Homo sapiens* | Amazonas, Brazil |
| 1561 | TcV | *T. cruzi* NR cl3 | *Homo sapiens* | Santa Cruz, Bolivia |
| 2121 | | *T. cruzi* 92:80 cl2 | *Homo sapiens* | Santa Cruz, Bolivia |
| 476 | TcVI | *T. cruzi* cl14 | *Triatoma infestans* | Rio Grande do Sul, Brazil |
| 2189 | | *T. cruzi* CL Brener cl1 | *Triatoma infestans* | Rio Grande do Sul, Brazil |
| 1994 | Tcbat | *T. cruzi* bat cl1.1 | *Myotis levis* | São Paulo, Brazil |
| 344 | | *T. c. marinkellei* | *Carollia persipicillata* | Rondônia, Brazil |
| 211 | | *T. dionisii* | *Epitesicus brasiliensis* | São Paulo, Brazil |
| 1946 | | *T. erneyi* | *Mops condylurus* | Sofala, Mozambique |
| 86 | | *T rangeli* AM80 | *Homo sapiens* | Amazonas, Brazil |

*TCC Trypanosomatid Culture Collection, Department of Parasitology, University of São Paulo.*

exclusively in *T. c. marinkellei*. Superoxide dismutase is a metabolic enzyme that protects the parasites from damage by oxidative stress, and is among proteins shown to have chemotherapeutic promise[86]. Structural proteins including paraflagellar rod proteins were also exclusively identified in *T. c. marinkellei* (K2LTU6) and *T. rangeli*. A Tubulin alpha chain (R9TNG7/A0A061J3L8/K2N695/V5AC55) and two paraflagellar rod 2C (A0A3R7KDB9 and A0A3S5IRI8) proteins were also identified exclusively in *T. rangeli*. Based on 2D-DIGE coupled to mass spectrometry, α-tubulins were identified as exclusive proteins for TcII and TcIV, while β-tubulins were identified exclusive to TcI and TcIII[56].

In conclusion, the PhyloQuant approach to profile and infer evolutionary relationships of organisms based on large scale quantitative mass spectrometry intensities was described, and validated by the high resolution of the DTUs harboring the highly heterogenetic *T. cruzi* strains, and their closest related trypanosome species from bats. This approach showed high-congruence to well-established phylogenetic clustering. Importantly, the phylogenetic character mapping of the evolutionary trees generated by the PhyloQuant approach allowed the identification of proteins and their expression profiles which support the different *T. cruzi* DTUs and the distinction of all closely related trypanosome species.

## Methods

**Trypanosome culture**. Two representative strains for the six human-infective *T. cruzi* DTUs (TcI-TcVI), one from Tcbat (TcVII), *T. cruzi marinkellei*, *Trypanosoma dionisii*, *Trypanosoma erneyi* and *Trypanosoma rangeli* (Table 1) were comparatively analyzed in this study. These parasites have been previously characterized using genetic markers[35,61]. The epimastigotes were grown in liver infusion tryptose medium supplemented with 10% fetal calf serum at 28 °C. The cells were harvested at the exponential growth phase by centrifugation at 3000 × g for 10 min and washed twice with ice cold PBS (137 mM NaCl, 2.7 mM KCl, 10 mM $Na_2HPO_4$, 1.8 mM $KH_2PO_4$, pH 7.4). Three biological replicates of epimastigote cultures were acquired for each trypanosome to guarantee the reproducibility of the results.

**Protein extraction and digestion**. The cell pellets were resuspended in protein extraction buffer containing 8 M urea and 1× protease inhibitor cocktail. The cells were lysed by three cycles of probe tip sonication on ice for 15 s with 15 s cooling intervals. The lysates were briefly vortexed and spun down, and the extracted proteins quantified by Quibit fluorometric detection method. Ammonium bicarbonate was added to a final concentration of 50 mM, followed by reduction using dithiothreitol (DTT) at a final concentration of 10 mM. The reactions were incubated for 45 min at 30 °C, and alkylated by the addition of iodoacetamide to a final concentration of 40 mM and incubation for 30 min in the dark at room

temperature. DTT at a final concentration of 5 mM was added to quench the reaction, and the urea diluted ten times by the addition of $NH_4HCO_3$. Porcin trypsin was used to digestion of the proteins at a ratio of 1:50 (μg trypsin/μg protein), and the mixture incubated overnight at 30 °C. The resultant peptide mixtures were centrifuged at 10,000 × g for 10 min, and 20 μg for each sample desalted separately with an in-house C18 micro-column of hydrophilic–lipophilic-balanced solid phase extraction. The peptides were eluted with 100 μL of 50% (v/v) acetonitrile and 0.1% (v/v) trifluoroacetic acid (TFA).

**Nano LC-MS/MS analysis**. LC-MS/MS analysis was performed on an EASY-Spray PepMap® 50 cm × 75 um C18 column using an Easy nLC1000 nanoflow system coupled to Orbitrap Fusion Lumos mass spectrometer (Thermo Fischer Scientific, Waltham, MA, USA). The HPLC gradient was 5–25% solvent B (A = 0.1% formic acid; B = 100% ACN, 0.1% formic acid) in 74 min at a flow of 300 nL/min. The most intense precursors selected from the FT MS1 full scan (resolving power 120,000 at m/z 200) were quadrupole-isolated and fragmented by higher-energy collision dissociation (HCD) and detected in the dual-pressure linear ion trap with 30 as normalized collision energy. The MS1 scan range was between 375 and 1600 m/z, the AGC target was set to $5 × 10^5$, the MS2 ion count target was set to $5 × 10^4$, and the max injection time was 50 and 54 ms for MS1 and MS2, respectively. The dynamic exclusion duration was set to 45 s with a 10-ppm mass tolerance around the selected precursor and its isotopes. The maximum total cycle time was confined to 3 s[23]. All raw data have been submitted to PRIDE archive[87], project accession: PXD017228.

**Peptide and protein identification and quantification**. LC-MS/MS raw files were analyzed using MaxQuant v1.5.2.8 for identification and LFQ of proteins and peptides. Using the following parameters, MS/MS spectra were searched against the combined reference Uniprot proteome databases of *T. cruzi* CL Brener, *T. cruzi* Dm 28c, *T. cruzi* marinkellei B7 and *T. rangeli* (strains SC58 and AM80) (Released, March, 2020; 50,075 entries) and common contaminants protein database with a mass tolerance level of 4.5 ppm for MS and 0.5 Da for MS/MS. Enzyme specificity was set to trypsin with a maximum of two missed cleavages. Carbamidomethylation of cysteine (57.0215 Da) was set as a fixed modification, and oxidation of methionine (15.9949 Da), deamidation NQ (+0.9840 Da) and protein N-terminal acetylation (42.0105 Da) were selected as variable modifications. The minimum peptide length was set to seven amino acid residues. The 'match between runs' feature in MaxQuant which enables peptide identifications between samples based on their accurate mass and retention time was applied with a match time window of 0.7 min and an alignment time window of 20 min. All identifications were filtered in order to achieve a protein false discovery rate (FDR) <1%[23,57]. Proteins identified in the reverse database, contaminants and proteins identified only by site were excluded prior to performing statistical analysis and evolutionary inferences.

**Evolutionary resolution of Schizotrypanum lineage**. Protein expression profiles evidenced by bottom-up large scale mass spectrometry-based quantitative proteomic features were analyzed to discriminate *T. cruzi* DTUs and closely related trypanosomes, and to infer the evolutionary relationships of the highly phylogenetically related trypanosomes in the subgenus *Schizotrypanum*. *T. rangeli* was included as the outgroup.

The m/z values of the parent ions (MS1) are unidentified features which are independent of database search, peptide identification and genome information. The MS1 intensities in the matched features file were filtered to include all peptide intensities with a charge state of ≥2. The cutoff of charge state 1 served to differentiate the majority of peptide ions from non-peptide ions which are single-charged. The filtered MS1 intensity values were normalized based on the total MS1 intensities for each replicate and processed further in the Perseus computational platform version 1.6.10.43[88]. The m/z peptide ion intensities were transformed to log2 values, and filtered based on three valid values in at least one replicate. The NaN were imputed from normal distribution using a down-shift of 1.8 and distribution width of 0.3. To visualize the variability of the MS1 intensities, PCA was performed on the log2-transformed MS1 intensities. Statistically significant MS1 intensities were determined by analysis of variance (ANOVA) with Benjamini–Hochberg-based FDR correction at an FDR <0.05. To infer evolutionary relationships between the trypanosome species based on MS1 intensities, total and statistically regulated MS1 intensities were normalized by z score transformation (standard deviations from means) in Perseus. To build character-based matrix for MP method, the proteomic matrix was transposed to present operational taxonomic units (OTUs) in rows and the intensities in columns. Subsequently, z scores were transformed into character states by rounding decimal fractions to the nearest integers, and the negative integers were substituted by corresponding letters. Finally, Tree Analyses using New Technology (TNT)[89] version 1.5[90] was employed to infer evolutionary relationships using MP. Branch statistical supports were obtained as implemented in TNT using 1000 replicates. The second quantitative measure used in this study was the iBAQ. This is a measure of absolute protein amounts calculated as the sum of all peptide peak intensities divided by the number of theoretically observable tryptic peptides[91]. The iBAQ values generated by MaxQuant software were normalized based on the total intensities for each replicate, loaded onto Perseus, and PCA used to visualize the variability of the proteins based on iBAQ intensities. Statistically significant iBAQ intensities were determined by ANOVA with Benjamini–Hochberg-based FDR correction at an FDR <0.05. To infer evolutionary relationships based on total and statistically significant iBAQ values, MP as implemented in TNT was used as previously described. Evolutionary trees were constructed based on total and statistically significant iBAQ values.

Thirdly, proteome-wide quantitative proteomics based on LFQ of fragmented ions (MS2) were evaluated to determine the discriminatory power of the total and significantly expressed proteins between the strains and species, and to reconstruct the evolutionary relationships of the trypanosomes. The LFQ values were analyzed by Perseus as previously described. To determine proteins with significant changes in abundances, ANOVA was applied with Benjamini–Hochberg-based FDR correction at an FDR <0.05. The total and significantly expressed protein LFQs were normalized using the z score and the evolutionary relationships inferred using parsimony as previously described. Phylogenetic character mapping[92] of the proteins supporting different clades, and their expression profiles, were inferred using the TNT program.

**Phylogenetic clustering**. Three genes; gGAPDH, V7V8 variable region of the SSU (small subunit) rRNA and heat shock protein-70 (HSP70) were selected and concatenated to construct a phylogenetic reference tree. These gene sequences have been routinely used to discriminate and infer phylogenetic relationships among trypanosomatids in general, resulting in the well-resolved phylogenetic inferences of *Schizotrypanum* trypanosomes and *T. rangeli*[61]. Orthologs of the three genes were obtained from the available genomes (complete and drafts) using a standalone BLASTn (v2.2.31+). The genomes from *T. cruzi* (Sylvio X10), *T. cruzi* (CL Brener) and *T. cruzi* (Esmeraldo) are available in TriTrypDB[93]. *T. cruzi* (Y) genome is available in NCBI-GeneBank[94]. The genome drafts from *T. dionisii* (TCC211), *T. erneyi* (TCC1946), *T. rangeli* AM80 (TCC86), *T. cruzi marinkellei* (TCC344), *T. cruzi* G (TCC874) and *T. cruzi* M-6241 (TCC2123) have been generated in our laboratories within the ATOL (Assembly of Life, NSF-USA) and TCC-USP (Brazil) projects and were obtained as previously described[62,71]. Genes for *T. cruzi* MT3869 (TCC844), *T. cruzi* CANIII (TCC2124), *T. cruzi* Jose Julio (TCC85), *T. cruzi* 92:80 (TCC2121) were retrieved from NCBI repository using BLASTn.

SSU rRNA gene from *T. cruzi* CL Brener (TCC2189), gGAPDH genes from *T. cruzi* NR Cl1 (TCC1561) and *T. cruzi* Cl14 (TCC476), and HSP70 genes from *T. cruzi* 92:80 (TCC2121), *T. cruzi* NR Cl1 (TCC1561) and *T. cruzi* Cl14 (TCC476) were amplified by PCR as previously described[61,95,96], and subsequently sequenced. All gene sequences were aligned using Clustal Omega within the Seaview v.4.7 graphical tool software[97]. Manual refining of the alignment was performed prior to inferring phylogenies by the parsimony method using TNT. Branch statistical supports were obtained using bootstrap with 1000 replicates.

**Phylogenetic and PhyloQuant correlation**. Evolutionary trees obtained using PhyloQuant and phylogenetics were converted into distance matrices using Patristic Distance Matrix and their correlation evaluated using the Mantel test using the Analysis of Phylogenetics and Evolution package of the R statistical language environment for statistical computing[98,99].

**Putative strain/DTU/species-specific protein identification**. Putative strain/DTU/species-specific proteins exclusively expressed in epimastigotes harvested in the exponential growth phase of the trypanosomes were determined using the average groups and numeric Venn diagram features implemented in Perseus computational platform. A cutoff of two valid values in at least one group for strain/species were considered for this analysis. Strain/species-specific proteins were represented on a bar graph. Proteins identified and quantified exclusively in the six human-infective DTUs (and not in other trypanosomes) were considered for subsequent analysis to determine specific DTU exclusive proteins. A cutoff of five valid values in at least 1 DTU was considered and mapped on a Venn diagram using Interactivenn[100].

**Statistics and reproducibility**. The trypanosome strains and statistical methods applied have been described in the relevant "Methods" section. The sample size was 17 (Table 1), with each sample analyzed in biological triplicates. One replicate each for *T. cruzi* Y strain (TCC1320) and *T. dionisii* (TCC211) were excluded from the final analysis. All PhyloQuant clustering results and statistical correlation can be reproduced according to the parameters described in the "Methods" section.

**Reporting summary**. Further information on research design is available in the Nature Research Reporting Summary linked to this article.

## Data availability

All mass spectrometry raw data have been submitted to PRIDE archive, project accession: PXD017228. The sequenced genes have been deposited in GenBank under the accession numbers MW345242, MW345243, MW345244, MW345245, MW345246 and MW325707.

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

## Acknowledgements

We acknowledge the use of the Willi Hennig Society edition of TNT. We are thankful to Prof. Erney Camargo for his constant support discussing the project and results, and Marta Campaner for trypanosome cultures. The National Science Foundation supported genome sequencing of some trypanosomes included in the present study (PI Gregory A. Buck, Virginia Commonwealth University, NSF DEB-0830056 Assembling the Tree of Life program: Phylum Euglenozoa). The work was supported by grants and fellowships from FAPESP (2014/06863-3, 2018/18257-1, 2018/15549-1, 2014/25494-9 to GP, 2018/13283-4 to GSDO and SNM (2017/04032-5)), CNPq (DOQ and "Productivity fellowship" to GP), CNPq (Universal Program and "Productivity fellowship" to MMGT) and postdoctoral fellowships from CAPES (AGCM and LRF).

## Author contributions

Conceptualization and experimental design: G.P. and S.N.M.; Methodology: S.N.M., A.G. C.-M., L.R.-F., G.S.d.O., C.M.F.R., D.Q., G.E.R., M.M.G.T. and G.P.; Data curation: S.N.M., A.G.C.-M., L.R.-F., G.S.d.O. and G.P.; Formal analysis: S.N.M., A.G.C.-M. and G.P.; Visualization: S.N.M., L.R.-F. and C.M.F.R; Funding acquisition: G.P.; Project administration: G.P.; Resources: G.E.R., G.P. and M.M.G.T; Software: S.N.M. and A.G.C.-M.; Supervision: G.P.; Writing-original draft: S.N.M., A.G.C.-M., M.M.G.T. and G.P.; Writing—review and editing: S.N.M., A.G.C.-M., L.R.-F., C.M.F.R., G.E.R., M.M.G.T. and G.P. All authors edited and approved the final manuscript.

## Competing interests

The authors declare no competing interests.

## Additional information

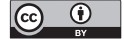

