## [Transparent Peer Review File · Communications Biology]

Reviewers' comments:

Reviewer #1 (Remarks to the Author):

Review of manuscript "New insights into *Trypanosoma cruzi* evolution and genotyping based on system-wide protein expression profiles (PhyloQuant)" submitted to Nature Communications

The authors describe in this manuscript a new method, PhyloQuant, for studying parasite evolution and genotyping using protein abundance measurements. The authors provide an extensive background of the parasite. However, as emphasized by the title, this is also a methods paper. As such, the authors should also provide a background of the field of using mass spectrometry in genotyping or phylogenetics. The authors do not cite the considerable literature on the evolutionary conservation of protein expression profiles (e.g. Schimpf et al. PLoS 2009) or the use of LC-MS/MS (including protein abundance information) in genotyping (e.g. Palmblad and Deelder, RCM 2012). Bacterial typing using mass spectrometry (including MALDI-TOF methods) were developed in the 1990s and were well established and commercialized before the cited 2015 paper.

The work itself appears solid and is clearly described (though with typos and omissions - see below), and this reviewer is in no way surprised to see this approach work, based on similar work described in the scientific literature. I believe the manuscript is too long given its content and novelty. The review of *T. cruzi* should be shortened.

Other comments:

Falling for what is essentially a vendor marketing trick from, the authors report the MS1 "resolution" (should be referred to as resolving power) at m/z 200, where it is higher. This number isn't directly relevant, as most tryptic peptides are observed as multiply charged species between m/z 500 and 1000. Indeed, in the very next sentence, the authors describe how the MS1 scan range was between 380 and 1500 m/z ! Reporting resolving power at m/z 200 in this context is about as relevant as citing the temperature in Australia in a US weather report.

The references are not correctly formatted and include random author affiliations in the list of author names, e.g.

981 95. Samir V. Deshpande^{1*} REJ, Peter A. Snyder², Michael Stanford², Charles H.
982 Wick² and Alan W. Zulich² (2011) ABOid: A Software for Automated Identification
983 and Phyloproteomics
984 Classification of Tandem Mass Spectrometric Data. J Chromatograph Separat Techniq
985 (5).

I could find this paper in PubMed using this information.

Reviewer #2 (Remarks to the Author):

The objective of this work was to apply a quantitative shotgun proteomics-based approach to analyze evolutionary relationships among different *T. cruzi* strains (DTUs) and among *T. cruzi* and other closely related trypanosomes. Global unidentified quantitative data (MS1) as well as differential protein expression profiles, were used to build phylogenetic trees used to compare for congruence with sequence-based phylogenies. The approach is interesting since it aims to explore evolutionary relationships among organisms from MS data and the authors propose the term "phyloquant" to name this approach. It should be noted that "Phyloproteomics" is a term that already exists in the literature to refer biological clustering from proteomic data. In that context, the term "phyloquant" is intended to encompass all "omics" studies, although this study focuses on proteomic data.

The approach and results of this work seems to be novel enough to be considered for publication in *Comm Biol*. However, this reviewer believes that the authors needs to address some substantive points before this manuscript can be accepted for publication. The authors need to re-write some parts and provide a more detailed analysis and explanations of some aspects. Please see below my points.

1) The results concerning "phyloquant" phylogenies, shown in Figure 3 and supplementary Figures 1 and 2, require further explanation. Phylogenetic trees were inferred by maximum parsimony (MP) from quantitative proteomic data. Authors need to explain how protein expression level data (eg intensity values) were used to construct trees and how changes of protein expression levels are represented on the tree branches. Clearly, they cannot be represented by substitutions per site, as stated in these figures. The authors should provide a more thorough explanation of these aspects. In general, I have difficulties to grasp the notion of how a character based phylogenetic approach, like MP, can be implemented with distance/similarity matrices.

2) There should also be further justification for the use of proteomic data for evolutionary inferences. What would be the advantage over the use of sequence (nucleotide or amino acid) data? This aspect is crucial as a central part of the justification of the work.

3) Correlation analysis of protein expression and divergence (Supplementary Figures 3-6) is hard to understand. How the authors measure divergence? The authors claim that they used *T. rangeli* as the most recent common ancestor to the *Schizotrypanum* clade (see lines 553-554) for analysis. How come? Being *T. rangeli* an existing species, in any case it shares a recent common ancestor with the clade. But *T. rangeli* itself is not a common ancestor. Furthermore, it is supposed that, for analysis, the authors may measure, for each protein represented in the correlograms, rate of evolution rather than evolutionary distance. The way in which these estimates have been made, as well as the meaning of the divergence lines shown in the figures, must be clarified. Finally, although they are presented as supplementary material, no interpretation of the results obtained is found in the discussion section of the work. The authors must elaborate and provide some analysis of these results or remove this part.

4) It is difficult to visualize the overall results regarding differentially expressed proteins when species/strains/DTUs are compared. The authors may wish to provide the results in such a way (for example as Venn diagrams) that it can be appreciated the number of common or exclusive proteins identified in this study comparing the different samples. This information is not easy to find in the study. However, it is important because it would facilitate the appreciation of the similarities and differences between the different groups. Likewise, information on the total number of proteins identified for each strain or species could be added in Table 2. This would allow us to appreciate the proteome coverage achieved for each sample.

5) The identification of unique peptides in order to characterize putative strain/DTU proteins revealed in most of the cases, members of multigene families. Do the authors have any explanation for this bias?

Regarding the species-specific proteins identified, it is striking that *T. rangeli* has such a low number of specific proteins (Table 2), being the most biologically different species. How do the authors explain this observation? Could there be a bias due to the presence of numerous missing values in their MS data?

6) Several terms are used throughout the manuscript to refer to the use of quantitative proteomic data to study evolutionary relationships. The authors propose the term phyloquant, as a more comprehensive term, to refer to studies of evolutionary inferences from data generated by MS. However, in various parts of the work the authors use the term phylomics and it is not understood what the difference is when using one or the other term. Authors should simplify the use of these terms.

7) The use of MS1 data seems to be adequate in this case in which a considerable number of samples are studied. This mitigates the missing data problem and improves quantitative accuracy and precision in the analysis including more low abundance species. However, it is unclear why only the most intense MS1 values are used.

Furthermore, throughout the work the authors use different features (MS1, iBAQ and LFQ) that in some cases give different results. What is its significance? How do the authors interpret these different results in their clustering studies? This reviewer believes that it is necessary to give a discussion on the different approaches used in this work.

8) The authors refer to different methods to construct phylogenetic trees (please see lines 575-577), but they did not explain why they chose maximum parsimony in this study and did not compare to the other approaches. Therefore, it is unclear why they mention these methods.

Some additional comments and format issues:

Line 40: replace "persons" with "people"

Line 45: consider include "intracellular" in the sentence "The amastigotes replicate..."

Line 46: replace "rapture" with "rupture"

Line 64: replace "genotypes" with "genotype"

Line 71: "erneyi" should be written in italics

Line 72: better write "T. cruzi clade" instead "clade T. cruzi"

Line 77: replace "Quantphylo" with "Phyloquant"

Line 78: to be consistent consider replace "ssrRNA" with "SSU rRNA" (see line 200)

Line 79: "levels of their levels of correlation" doesn't sound good

Line 206: replace "TryTrypDB" with "TriTrypDB" and "NCBI-Genenank" with "NCBI-GenBank"

Line 213: consider write "CL Brener" instead "Cl Brener"

Line 222: "phylomics" As already said, that word is not a term that has been introduced in the work, I suggest replacing it with "phyloproteomics"

Line 238: consider replace "family" with "families"

Line 265: delete "proteins"

Line 277: "normalize by z-square", did you mean z-score by using Perseus platform?

Line 305: replace "clusteres" with "clusters"

Line 328: replace "ssu rRNA" with "SSU rRNA" and "HSP" with "HSP70"

Line 352: correct "phyloquant"

Line 361: correct "hybrid"

Line 362: correct "genotypes" and "harboring"

Line 373: to be consistent consider replace "ssrRNA" with "SSU rRNA"
Line 416: "A total of 18 proteins..." according to table should be 17
Line 420: replace "showed that it's" with "showed its"
Lines 445-448: It is established a link between DTU TcII and benznidazole resistance but, is there experimental evidence that the 3869 strain is resistant to bz?
Line 446: replace "893" with "3869" and "were" with "where"
Line 451: correct "Glucose-6-phosphate"
Line 461: in "was identified with in..." delete "with"
Line 552: "T. cruzi" should be written in italics
Line 563: "Schizotrypanum" should be written in italics
Line 569: consider replace "to infer methodologies" with "to refer methodologies"
Line 571: replace "glycomis" with "glycomics"
Line 585: delete "jejuni"
Line 643: reference of Telleria et al. should be 2010 instead 2011
Line 661: correct "harboring"
Lines 708 and 710: "T. cruzi" should be written in italics
Line 1075: correct "Schizotrypanum"
Line 1080: replace "LFF" with "LFQ"
Lines 1089, 1099, 1110, 1121: replace "eclipses" with "ellipses"
Page 31, Figure 1: correct "epimastigote", and replace "stage" with "phase"

Response to reviewers

We thank reviewer #1 for the comments and suggestions to this study, which have contributed to improving the quality of the manuscript. In particular, we have included pioneer studies which described the application of mass spectrometry in genotyping. In addition, we have greatly summarized the literature on *T. cruzi*, and focused more on the PhyloQuant method development. We feel that this has greatly improved the revised manuscript.

The comments from, and responses to reviewer #1 are detailed below:

Reviewer 1

The authors describe in this manuscript a new method, PhyloQuant, for studying parasite evolution and genotyping using protein abundance measurements. The authors provide an extensive background of the parasite. However, as emphasized by the title, this is also a methods paper. As such, the authors should also provide a background of the field of using mass spectrometry in genotyping or phylogenetics. The authors do not cite the considerable literature on the evolutionary conservation of protein expression profiles (e.g. Schrimpf et al. PLoS 2009) or the use of LC-MS/MS (including protein abundance information) in genotyping (e.g. Palmblad and Deelder, RCM 2012). Bacterial typing using mass spectrometry (including MALDI-TOF methods) were developed in the 1990s and were well established and commercialized before the cited 2015 paper.

Reply:

We have revised our introduction and discussion, including early research by the different authors who pioneered the application of mass spectrometry in genotyping.

The work itself appears solid and is clearly described (though with typos and omissions - see below), and this reviewer is in no way surprised to see this approach work, based on similar work described in the scientific literature. I believe the manuscript is too long given its content and novelty. The review of *T. cruzi* should be shortened.

Reply:

All typos suggested by the reviewer have been corrected, and the omissions in the manuscript clarified and detailed in depth to facilitate complete understanding of the methodological steps and results obtained using the proposed PhyloQuant approach. In addition, we have summarized the literature on *T. cruzi* in the discussion on

strain/DTU/species-specific proteins sections, calling more attention and focus on the PhyloQuant method development and subsequent application of this technique.

Other comments:

Falling for what is essentially a vendor marketing trick from, the authors report the MS1 "resolution" (should be referred to as resolving power) at m/z 200, where it is higher. This number isn't directly relevant, as most tryptic peptides are observed as multiply charged species between m/z 500 and 1000. Indeed, in the very next sentence, the authors describe how the MS1 scan range was between 380 and 1500 m/z ! Reporting resolving power at m/z 200 in this context is about as relevant as citing the temperature in Australia in a US weather report.

Reply:

We have modified this section, replacing the word resolution for resolving power. We agree with the reviewer on reporting the resolving power at m/z 200; however, we have left this information in order to accurately describe the instrument settings defined by the company for better reproducibility of the experiment in other laboratories using the same instrument.

The references are not correctly formatted and include random author affiliations in the list of author names, e.g.

95. Samir V. Deshpande^{1*} REJ, Peter A. Snyder², Michael Stanford², Charles H. Wick² and Alan W. Zulich² (2011) ABOid: A Software for Automated Identification and Phyloproteomics984 Classification of Tandem Mass Spectrometric Data. J Chromatograph Separate Techniq (5). I could find this paper in PubMed using this information.

Reply:

All the references in the manuscript have been corrected.

Reviewer 2

We would like to thank reviewer #2 for the suggestions and comments to our study. The resulting clarifications have helped improve the quality of the manuscript presented in the revised version, which are highlighted below. Particularly, 1) we have highlighted the merits of the PhyloQuant approach compared to sequence-based phylogenetics, including the mapping of characters to the trees (synapomorphies) which support the different clades. To this end, we have added **Supplementary Table 4**, showing the proteins and their expression values which support the different DTU clades (synapomorphies). The information from synapomorphies is missing from sequence based phylogenetics, and offers information to understand and potentially link the genetic heterogeneity of *T. cruzi* DTUs from the potential correlation to the variable Chagas disease outcomes. 2) We have re-written and clarified some parts of the manuscript and figures (**Figure 1**) which were not detailed enough to allow a complete understanding of the proposed Phyloquant methodological steps applied in this study.

The comments from, and responses to reviewer #2 are detailed below:

The objective of this work was to apply a quantitative shotgun proteomics-based approach to analyze evolutionary relationships among different *T. cruzi* strains (DTUs) and among *T. cruzi* and other closely related trypanosomes. Global unidentified quantitative data (MS1) as well as differential protein expression profiles, were used to build phylogenetic trees used to compare for congruence with sequence-based phylogenies. The approach is interesting since it aims to explore evolutionary relationships among organisms from MS data and the authors propose the term “phyloquant” to name this approach. It should be noted that “Phyloproteomics” is a term that already exists in the literature to refer biological clustering from proteomic data. In that context, the term “phyloquant” is intended to encompass all “omics” studies, although this study focuses on proteomic data.

The approach and results of this work seems to be novel enough to be considered for publication in Comm Biol. However, this reviewer believes that the authors needs to address some substantive points before this manuscript can be accepted for publication. The authors need to re-write some parts and provide a more detailed analysis and explanations of some aspects. Please see below my points.

Question 1

The results concerning “phyloquant” phylogenies, shown in **Figure 3** and supplementary Figures 1 and 2, require further explanation. Phylogenetic trees were inferred by maximum parsimony (MP) from quantitative proteomic data. Authors need to explain how protein expression level data (eg intensity values) were used to construct trees and how changes of protein expression levels are represented on the tree branches. Clearly, they cannot be represented by substitutions per site, as stated in these figures. The authors should provide a more thorough explanation of these aspects. In general, I have difficulties to grasp the notion of how a character based phylogenetic approach, like MP, can be implemented with distance/similarity matrices

Reply:

The methodological steps of the proposed ‘*PhyloQuant*’ approach have been clarified (in the Materials and Methods section and in **Figure 1**) to enable the complete understanding of the proposed method of inferring evolutionary relationships from protein expression values. The analytical workflow has been detailed to illustrate the step-by-step analysis from protein expression values/intensities to phylogenetic trees generated by Parsimony analysis using TNT program.

In addition, we have changed how the expression levels are represented on the tree branches, replacing ‘substitutions per site’ to ‘protein expression changes’.

Question 2.

There should also be further justification for the use of proteomic data for evolutionary inferences. What would be the advantage over the use of sequence (nucleotide or amino acid) data? This aspect is crucial as a central part of the justification of the work.

Reply:

The use of proteomic data and especially protein expression levels to infer evolutionary relationships offers added information to sequence based phylogenies. 1) For instance, the genome of *T. cruzi marinkelli* revealed that *T. c. marinkellei* and *T. c. cruzi* are very similar, with the exception of one gene, an acetyltransferase gene, which was identified only in *T. c. marinkellei*. Indeed, the distinction of *T. cruzi* and *T. cruzi marinkellei* to infect mammalian hosts can thus be related to the differential protein expression between the parasites and not the genome make-up.

2) In addition, the virulence of the different *T. cruzi* DTUs can be attributed to the differential expression of a protein or group of proteins, information which is lacking in gene-based classifications. Different *T. cruzi* DTUs have been reported to be associated with different clinical outcomes and classification based on protein expression can allow us to infer more information on this aspect. The PhyloQuant approach can be applied also to different *Leishmania* species which have highly similar genomes, but cause very different diseases, from cutaneous/tegumentary to visceral leishmaniasis. The classification of *Leishmania* species based on differential protein expression, and the identification of proteins which support the different clades, can help elucidate the parasite-host interaction and the disease outcome.

3) The application of protein expression profiles to study evolutionary relationships allows the mapping of the expression of protein families (synapomorphies) supporting the tree branches. We have added **Supplementary Table 4** showing the proteins that support the different trypanosomatids clades (e.g. proteins supporting TcI-TcBat sister clade, proteins supporting parental and hybrid genotypes, proteins supporting the clade separating *Schizotrypanum* clade from *T. rangeli*, etc), and their differential expression profiles.

4) In addition, this approach helps to uncover events of clade specific adaptations and convergent evolution between clades or species.

The merits of the Phyloquant have been discussed in the revised draft

Question 3

Correlation analysis of protein expression and divergence (Supplementary Figures 3-6) is hard to understand. How the authors measure divergence? The authors claim that they used *T. rangeli* as the most recent common ancestor to the Schizotrypanum clade (see lines 553-554) for analysis. How come? Being *T. rangeli* an existing species, in any case it shares a recent common ancestor with the clade. But *T. rangeli* itself is not a common ancestor. Furthermore, it is supposed that, for analysis, the authors may measure, for each protein represented in the correlograms, rate of evolution rather than evolutionary distance. The way in which these estimates have been made, as well as the meaning of the divergence lines shown in the figures, must be clarified. Finally, although they are presented as supplementary material, no interpretation of the results obtained is found in the discussion section of the work. The authors must elaborate and provide some analysis of these results or remove this part.

Reply:

Following the reviewer's suggestion, we have removed this part.

Question 4

It is difficult to visualize the overall results regarding differentially expressed proteins when species/strains/DTUs are compared. The authors may wish to provide the results in such a way (for example as Venn diagrams) that it can be appreciated the number of common or exclusive proteins identified in this study comparing the different samples. This information is not easy to find in the study. However, it is important because it would facilitate the appreciation of the similarities and differences between the different groups. Likewise, information on the total number of proteins identified for each strain or species could be added in **Table 2**. This would allow us to appreciate the proteome coverage achieved for each sample.

Reply:

We have included the total and regulated proteins identified in this study in **Supplementary Table 2** to infer the total number of proteins identified for each strain/species.

We have presented the strain/species-specific proteins in **Figure 4A** in a bar graph, indicating the number of proteins identified exclusively for each strain/species. The strain/species specific proteins have been included as **Supplementary Table 5**.

In addition, the exclusive human infective DTU-specific proteins identified in this study have been represented as a Venn diagram (**Figure 4B**), as suggested by the reviewer, and are presented as **Supplementary Table 5**. The total number of proteins identified in each human infective DTUs are highlighted in parenthesis in the Venn diagram.

The section on strain/DTU/species-specific proteins has been extensively reconstructed and summarized, shortening the manuscript.

Question 5

The identification of unique peptides in order to characterize putative strain/DTU proteins revealed in most of the cases, members of multigene families. Do the authors have any explanation for this bias?

Regarding the species-specific proteins identified, it is striking that *T. rangeli* has such a low number of specific proteins (Table 2), being the most biologically different species. How do the authors explain this observation? Could there be a bias due to the presence of numerous missing values in their MS data?

Reply:

The identification of multigene families could be associated with their diversity in expression compared to single copy genes. This diversity might represent a unique signature for each strain/DTU interacting with its hosts in a differential manner. The multigene families constitute 50% of the total genome and their content has been shown to be different in phylogenetically distant strains. Their differential expression profile in different DTUs might regulate the infectivity, host immune response, cellular proliferation and transmissions (PMID: 22431647). These features have been reported to be associated with particular *T. cruzi* strains located in a selected geographical region.

We have researched all raw data using a newer version of the Uniprot reference proteome databases (*T. cruzi* CL Brener, *T. cruzi* Dm28c, *T. c. marinkellei* and *T. rangeli*). This information has been included in the materials and methods. Using this database, the number of *T. rangeli* unique proteins is the largest one as expected being the most biologically different species within the ones compared.

Question 6

Several terms are used throughout the manuscript to refer to the use of quantitative proteomic data to study evolutionary relationships. The authors propose the term phyloquant, as a more comprehensive term, to refer to studies of evolutionary inferences from data generated by MS. However, in various parts of the work the authors use the term phylomics and it is not understood what the difference is when using one or the other term. Authors should simplify the use of these terms.

Reply:

The term 'phylomics' has been replaced with 'PhyloQuant' in the text.

Question 7

The use of MS1 data seems to be adequate in this case in which a considerable number of samples are studied. This mitigates the missing data problem and improves

quantitative accuracy and precision in the analysis including more low abundance species. However, it is unclear why only the most intense MS1 values are used.

Furthermore, throughout the work the authors use different features (MS1, iBAQ and LFQ) that in some cases give different results. What is its significance? How do the authors interpret these different results in their clustering studies? This reviewer believes that it is necessary to give a discussion on the different approaches used in this work.

Reply:

Based on the reviewer's suggestion we have inferred evolutionary relationships using the total and statistically significant MS1 features. Based on the Mantel test, trees generated using both MS1 feature extraction methods were statistically similar to phylogenetic trees.

Moreover, we wanted to compare different quantitative features extractable from these data to test the PhyloQuant approach. Each method was described in detail in the "Materials and Methods" section. Using different MS quantitative data (MS1, iBAQ and LFQ) resulted in similar topologies with minor, not significant, differences. All the evolutionary relationships were maintained using the three quantitative MS features. These differences were not significant, keeping all the tree topologies consistent with the phylogenetic distances. As a comparison with phylogenetic approaches, different gene sequences generate alternative tree topologies keeping intact the phylogenetic distances between strains and species. The PhyloQuant approach using MS1, iBAQ or LFQ performed with similar quality and reproducibility to infer evolutionary relationships.

Question 8

The authors refer to different methods to construct phylogenetic trees (please see lines 575-577), but they did not explain why they chose maximum parsimony in this study and did not compare to the other approaches. Therefore, it is unclear why they mention these methods.

Reply:

Maximum parsimony (MP) method of evolutionary studies uses the lowest number of evolving steps to drive the search for the evolutionary trees. MP is a simple method which does not need evolutionary models for clustering, as maximum likelihood (ML) and Bayesian inference (BI) do. In addition, MP can track on the tree the evolving steps, meaning that it is able to map the different proteins that support a clade

(synapomorphies), unlike distance methods and neighbor joining methods. We have added **Supplementary Table 4** showing the proteins that support the different trypanosomatids clades (e.g. protein expression profiles supporting TcI-TcBat sister clade, proteins supporting parental and hybrid genotypes, proteins supporting the clade separating *Schizotrypanum* clade from *T. rangeli*, etc). Moreover, the merits of using MP over the other clustering methods have been included in the revised version of this manuscript.

Some additional comments and format issues:

Line 40: replace “persons” with “people”

“Persons” has been replaced with “people” (Line 40)

Line 45: consider include “intracellular” in the sentence “The amastigotes replicate...”

“intracellularly” has been inserted (Line 50)

Line 46: replace “rapture” with “rupture”

Rapture has been replaced with “rupture” (Line 52)

Line 64: replace “genotypes” with “genotype”

This phrase was reformatted (Line 59)

Line 71: “erneyi” should be written in italics

“erneyi” has been italicized (Line 87)

Line 72: better write “T. cruzi clade” instead “clade T. cruzi”

“clade T. cruzi” has been replaced with “T. cruzi clade” (Line 87)

Line 77: replace “Quantphylo” with “Phyloquant”

Quantphylo has been replaced with “PhyloQuant” (Line 93)

Line 78: to be consistent consider replace “ssrRNA” with “SSU rRNA” (see line 200)

“ssrRNA” has been replaced with “SSU rRNA” (Line 94)

Line 79: “levels of their levels of correlation” doesn’t sound good

This phrase has been formatted to read “to determine their levels of correlation” (Line 95)

Line 206: replace “TryTrypDB” with “TriTrypDB” and “NCBI-Genenank” with “NCBI-GenBank”

Corrected to “TriTrypDB” and “NCBI-GeneBank” (Line 216)

Line 213: consider write “CL Brener” instead “Cl Brener”

CL Brener has been corrected (Line 222)

Line 222: “phylomics” As already said, that word is not a term that has been introduced in the work, I suggest replacing it with “phyloproteomics”

Phylomics has been replaced with PhyloQuant (Line 230)

Line 238: consider replace “family” with “families”

The section on Correlation with protein families has been removed from the revised draft

Line 265: delete “proteins”

Corrected (Line 264)

Line 277: “normalize by z-square”, did you mean z-score by using Perseus platform?

“normalize by Z-score” has been corrected (Line 268)

Line 305: replace “clusteres” with “clusters”

Corrected

Line 328: replace “ssu rRNA” with “SSU rRNA” and “HSP” with “HSP70”

Corrected to SSU rRNA and HSP70 (Line 370)

Line 352: correct “phyloquant”

Corrected to “PhyloQuant” (Line 368)

Line 361: correct “hybrid”

Corrected to “hybrid” (Line 357)

Line 362: correct “genotypes” and “harboring”

Changed (Line 356)

Line 373: to be consistent consider replace “ssrRNA” with “SSU rRNA”

Corrected (Line 370)

Line 416: “A total of 18 proteins...” according to table should be 17

Corrected to 17 in Figure 4A. The text on the Strain/Species and DTU specific proteins has been reformatted and shortened (Line 399).

Line 420: replace “showed that it’s” with “showed its”

This section of the draft was reformatted

Lines 445-448: It is established a link between DTU TcII and benznidazole resistance but, is there experimental evidence that the 3869 strain is resistant to bz?

We agree with the reviewer that no data is available on bz resistance in *T. cruzi* strain 3869. We have reformatted the phrase to represent this fact. (lines 585-593)

Line 446: replace “893” with “3869” and “were” with “where”

Corrected

Line 451: correct “Glucose-6-phosphate”

This section of the manuscript was deleted

Line 461: in “was identified with in...” delete “with”

This section of the draft was deleted

Line 552: “*T. cruzi*” should be written in italics

All “*T. cruzi*” in the text have been italicized.

Line 563: “*Schizotrypanum*” should be written in italics

All “*Schizotrypanum*” in the text have been italicized

Line 569: consider replace “to infer methodologies” with “to refer methodologies”

“infer methodologies” has been corrected to “to indicate methodologies” (Line 428)

Line 571: replace “glycomis” with “glycomics”

Glycomic has been corrected (Line 431)

Line 585: delete “jejuni”

Jejuni has been deleted (Line 451)

Line 643: reference of Telleria et al. should be 2010 instead 2011

Reference has been corrected (Line 491)

Line 661: correct “harboring”

“harboring” has been corrected to “harboring” (Lines 357 and 504)

Lines 708 and 710: “T. cruzi” should be written in italics

“T. cruzi” has been italicized

Line 1075: correct “Schizotrypanum”

Corrected (line 1024)

Line 1080: replace “LFF” with “LFQ”

LFF has been corrected to LFQ (Line 1029)

Lines 1089, 1099, 1110, 1121: replace “eclipses” with “ellipses”

The divergence analysis has been removed from the revised draft

Page 31, Figure 1: correct “epimastigote”, and replace “stage” with “phase”

Figure 1 has been corrected to “epimastigote” and “exponential growth phase”

REVIEWERS' COMMENTS:

Reviewer #1 (Remarks to the Author):

The authors have largely addressed the concerns of the reviewers. However, the manuscript reads more like a review than an article describing a new method.

It is great to see that the authors now refer to some of the earlier work on using mass spectrometry in molecular phylogenetics and species identification. However, they cite five papers on pyrolysis methods and no fewer than nine works on MALDI-TOF identification of bacteria. This would be fine in a review, but this is not a review article. One or two references to the first publication and/or a recent review would suffice. It would be more relevant to cite the early uses of mass spectrometry in phylogenetics and identification of eukaryotes. The three citations included are relevant, though 68 and 69 are slightly redundant and Wulff et al. 2013 an earlier and independent example of ID-free, genome independent proteomics workflow for eukaryote identification. The authors managed to find very early work on bacteria identification (e.g. by Fenselau et al.), and some interesting early work this reviewer was not aware of, suggesting they are passionate about the history and background of the field(s). I would therefore strongly recommend they remove the review aspects of this manuscript, along with most of the 80 (!!!) references on the parasite, and use all of this in one or two future review papers. This would allow them to focus on describing and demonstrating the new method in this article.

Reviewer #2 (Remarks to the Author):

This manuscript, in its revised version, addresses the comments and criticisms raised on its previous version. Consequently, the work has been extensively modified.

Here are some comments on the answers given by the authors,

1. The authors have clarified how they transform protein expression data to character-state changes and how expression profiles allowed them the mapping of synapomorphies. Figures 1 and 3 have been modified as suggested. I believe that this is an important part of the work, which has been included in this revised version providing valuable information about protein expression profiles that support the different clades.

2. It has been further justified and highlighted proteomics approach. It has also been clarified the contribution that this MS-based approach means for the study of phylogenetic relationships. It is clear in that sense, that the PhyloQuant approach is complementary and congruent to sequence-based clustering approaches and can discriminate well known phylogenetic relationships of DTUs and trypanosome species. This is demonstrated by the high correlation that the authors found between both approaches in this study.

The authors answered extensively, providing examples about the advantages of the Phyloquant approach in understanding the biological characteristics of different strains/species, but it should be noted that what was necessary to justify was its use for evolutionary inferences and not to identify differential protein expression profiles that could explain the different clinical outcomes. Clearly, the identification of differential profiles of protein expression can contribute to the understanding of parasite-host interactions and infection strategies of the different strains/species.

The authors said that *T. c. marinkellei* and *T. c. cruzi* genomes are remarkably similar, except for one

gene, an acetyltransferase gene, which was identified only in *T. c. marinkellei* (Frazen et al. BMC Genomics 2012, 13:531). But comparative genomic analysis of *T. c. marinkellei* and *T. c. cruzi* Sylvio X10 performed by Frazen et al. showed that these genomes differ in several aspects, e.g., copy number of various genes, showing mainly differences in the number of members of various multigene families. So, it would not be correct to say that these genomes differ by only one gene. What Frazen et al report is the identification of a single copy gene present in the *T. c. marikellei* genome and that is absent in both *T. c. cruzi* Sylvio X10 and *T.c.c. CL Brener*. This is different from saying that these genomes differ by a single gene.

According to this I recommend modifying lines 460-461.

3. Regarding the section that covered divergence analysis, this part, that needed to be extensively modified, has been removed, which was suggested as one of the alternatives.

4. As suggested, the authors have provided complete information on the total number of proteins and differentially expressed proteins identified for each sample (supplementary figures 2 and 5). According to that, figure 4 was added. So now, it can be appreciated the number of common or exclusive proteins identified in this study comparing the different samples. I think this has done a lot to improve the work.

5. The authors re-analyzed the MS data and corrected the number of specific proteins for *T. rangeli*. While the numbers of unique proteins in the different strains and species showed few variations with respect to the previous analysis, in the case of *T. rangeli* this number went from 4 to 462 which seems to be more in line with what is expected.

6. The terms have been simplified by using "PhyloQuant" in all the manuscript. This has improved the clarity of the text.

7. MS data was re-analyzed using total and statistically significant values. The authors explain that the comparison of the three quantitative MS features showed quite similar results.

8. It has been incorporated into the text a more consistent explanation about the use of maximum parsimony and explaining how a character-based phylogenetic approach such as MS allows mapping synapomorphies.

Finally, a few minor issues in this revised version

- line 244, "proteins", remove one of the words

- lines 457-458, the sentence "However, although infection..." is repeated on lines 462-463. Delete some of them

- line 605 correct genome s

Therefore, with some minor modifications, I consider the paper ready to be published.

REVIEWERS' COMMENTS:

We thank the reviewers for the second revision of the manuscript and their positive comments. The revised document has been greatly improved by the suggested revisions, which are listed below.

Reviewer #1 (Remarks to the Author):

The authors have largely addressed the concerns of the reviewers. However, the manuscript reads more like a review than an article describing a new method.

It is great to see that the authors now refer to some of the earlier work on using mass spectrometry in molecular phylogenetics and species identification. However, they cite five papers on pyrolysis methods and no fewer than nine works on MALDI-TOF identification of bacteria. This would be fine in a review, but this is not a review article. One or two references to the first publication and/or a recent review would suffice. It would be more relevant to cite the early uses of mass spectrometry in phylogenetics and identification of eukaryotes. The three citations included are relevant, though 68 and 69 are slightly redundant and Wulff et al. 2013 an earlier and independent example of ID-free, genome independent proteomics workflow for eukaryote identification. The authors managed to find very early work on bacteria identification (e.g. by Fenselau et al.), and some interesting early work this reviewer was not aware of, suggesting they are passionate about the history and background of the field(s). I would therefore strongly recommend they remove the review aspects of this manuscript, along with most of the 80 (!!!) references on the parasite, and use all of this in one or two future review papers. This would allow them to focus on describing and demonstrating the new method in this article.

Reply:

Early studies describing the application of mass spectrometry in typing have been summarized, citing the two papers suggested by Reviewer #1. This has greatly improved the introduction, and allowed more focused introduction on PhyloQuant method development.

Reviewer #2 (Remarks to the Author):

This manuscript, in its revised version, addresses the comments and criticisms raised on its previous version. Consequently, the work has been extensively modified.

Here are some comments on the answers given by the authors,

1. The authors have clarified how they transform protein expression data to character-state changes and how expression profiles allowed them the mapping of synapomorphies. Figures 1 and 3 have been modified as suggested. I believe that this is an important part of the work, which has been included in this revised version providing valuable information about protein expression profiles that support the different clades.

2. It has been further justified and highlighted proteomics approach. It has also been clarified the contribution that this MS-based approach means for the study of phylogenetic relationships. It is clear in that sense, that the PhyloQuant approach is complementary and congruent to sequence-based clustering approaches and can discriminate well known phylogenetic relationships of DTUs and

trypanosome species. This is demonstrated by the high correlation that the authors found between both approaches in this study.

The authors answered extensively, providing examples about the advantages of the PhyloQuant approach in understanding the biological characteristics of different strains/species, but it should be noted that what was necessary to justify was its use for evolutionary inferences and not to identify differential protein expression profiles that could explain the different clinical outcomes. Clearly, the identification of differential profiles of protein expression can contribute to the understanding of parasite-host interactions and infection strategies of the different strains/species.

Reply:

In cases where genomes or gene sequences are missing, the PhyloQuant approach, specifically MS1-based intensities, can be used to infer evolutionary inferences or typing without prior knowledge of genome and protein identification. In addition, the assignment of the correct genotype of T. cruzi clinical isolates can be performed using the PhyloQuant approach.

The authors said that *T. c. marinkellei* and *T. c. cruzi* genomes are remarkably similar, except for one gene, an acetyltransferase gene, which was identified only in *T. c. marinkellei* (Frazen et al. BMC Genomics 2012, 13:531). But comparative genomic analysis of *T. c. marinkellei* and *T. c. cruzi* Sylvio X10 performed by Frazen et al. showed that these genomes differ in several aspects, e.g., copy number of various genes, showing mainly differences in the number of members of various multigene families. So, it would not be correct to say that these genomes differ by only one gene. What Frazen et al report is the identification of a single copy gene present in the *T. c. marinkellei* genome and that is absent in both *T. c. cruzi* Sylvio X10 and *T.c.c. CL Brener*. This is different from saying that these genomes differ by a single gene.

According to this I recommend modifying lines 460-461.

Modified accordingly (lines 261-262), expounding on the T. c. marinkellei exclusive gene sequences including the acetyltransferase gene reported by Frazen and colleagues (2012), alongside variations in coding and non-coding gene copy numbers, gene families and nucleotide polymorphisms .

3. Regarding the section that covered divergence analysis, this part, that needed to be extensively modified, has been removed, which was suggested as one of the alternatives.

4. As suggested, the authors have provided complete information on the total number of proteins and differentially expressed proteins identified for each sample (supplementary figures 2 and 5). According to that, figure 4 was added. So now, it can be appreciated the number of common or exclusive proteins identified in this study comparing the different samples. I think this has done a lot to improve the work.

5. The authors re-analyzed the MS data and corrected the number of specific proteins for *T. rangeli*. While the numbers of unique proteins in the different strains and species showed few variations with respect to the previous analysis, in the case of *T. rangeli* this number went from 4 to 462 which seems to be more in line with what is expected.

6. The terms have been simplified by using “PhyloQuant” in all the manuscript. This has improved the clarity of the text.

7. MS data was re-analyzed using total and statistically significant values. The authors explain that the comparison of the three quantitative MS features showed quite similar results.

8. It has been incorporated into the text a more consistent explanation about the use of maximum parsimony and explaining how a character-based phylogenetic approach such as MS allows mapping synapomorphies.

Finally, a few minor issues in this revised version

- line 244, “proteins”, remove one of the words

Corrected

- lines 457-458, the sentence “However, although infection...” is repeated on lines 462-463. Delete some of them

Corrected

- line 605 correct genome s

Corrected